# Modulation of T Regulatory and Dendritic Cell Phenotypes Following Ingestion of *Bifidobacterium longum*, AHCC^®^ and Azithromycin in Healthy Individuals

**DOI:** 10.3390/nu11102470

**Published:** 2019-10-15

**Authors:** Abeed H. Chowdhury, Miguel Cámara, Chandan Verma, Oleg Eremin, Anil D. Kulkarni, Dileep N. Lobo

**Affiliations:** 1Gastrointestinal Surgery, Nottingham Digestive Diseases Centre, National Institute for Health Research (NIHR) Nottingham Biomedical Research Centre, Nottingham University Hospitals NHS Trust and University of Nottingham, Queen’s Medical Centre, Nottingham NG7 2UH, UK; abeedchowdhury@aol.com (A.H.C.);; 2School of Life Sciences, Centre for Biomolecular Sciences, University of Nottingham, Nottingham NG7 2RD, UK; 3Department of Surgery, The University of Texas Health Science Center and McGovern Medical School, 6431 Fannin Street, MSB 4022-B, Houston, TX 77030, USA; Anil.D.Kulkarni@uth.tmc.edu; 4MRC Versus Arthritis Centre for Musculoskeletal Ageing Research, School of Life Sciences, University of Nottingham, Queen’s Medical Centre, Nottingham NG7 2UH, UK

**Keywords:** probiotics, prebiotics, synbiotics, antibiotics, dendritic cells, immunity, *Bifidobacterium longum*, AHCC^®^

## Abstract

The probiotic Bifidus BB536 (BB536), which contains *Bifidobacterium longum*, has been shown to have enhanced probiotic effects when given together with a standardized extract of cultured *Lentinula edodes* mycelia (AHCC^®^, Amino Up Co. Ltd., Sapporo, Japan). BB536 and AHCC^®^ may modulate T cell and dendritic cell (DC) phenotypes, and cytokine profiles to favour anti-inflammatory responses following antibiotic ingestion. We tested the hypothesis that orally administered BB536 and/or AHCC^®^, results in modulation of immune effector cells with polarisation towards anti-inflammatory responses following antibiotic usage. Forty healthy male volunteers divided into 4 equal groups were randomised to receive either placebo, BB536, AHCC^®^ or a combination for 12 days in a double-blind manner. After 7 days volunteers also received 250 mg azithromycin for 5 days. Cytokine profiles from purified CD3+ T cells stimulated with PDB-ionomycin were assessed. CD4+ CD25+ forkhead box P3 (Foxp3) expression and peripheral blood DC subsets were assessed prior to treatment and subsequently at 7 and 13 days. There was no difference in cytokine secretion from stimulated CD3+ T cells between treatment groups. Compared with baseline, Foxp3 expression (0.45 ± 0.1 vs. 1.3 ± 0.4; *p* = 0.002) and interferon-gamma/interleukin-4 (IFN-γ/IL-4) ratios were increased post-treatment in volunteers receiving BB536 (*p* = 0.031), although differences between groups were not significant. For volunteers receiving combination BB536 and AHCC^®^, there was an increase in myeloid dendritic cells (mDC) compared with plasmacytoid DC (pDC) counts (80% vs. 61%; *p* = 0.006) at post treatment time points. mDC2 phenotypes were more prevalent, compared with baseline, following combination treatment (0.16% vs. 0.05%; *p* = 0.002). Oral intake of AHCC^®^ and BB536 may modulate T regulatory and DC phenotypes to favour anti-inflammatory responses following antibiotic usage.

## 1. Introduction

Administration of antibiotics, whilst valuable for the treatment of bacterial infections, is associated with adverse effects that include disruption of gastrointestinal ecology. This may allow potentially harmful bacteria to proliferate, cause gastrointestinal symptoms [1,2] and increase the rates of antibiotic resistance [3]. Strategies to reduce these effects include the use of probiotics.

Probiotics are characterised by their ability to survive the harsh acidic environment of the stomach and subsequent transit through the gut where they are known to impart favourable effects for the host, especially in conditions with significant intestinal dysbiosis [4,5]. Evidence for these effects comes from randomised studies suggesting that probiotics are useful along with rehydration therapy for the treatment of infectious diarrhoea in children [6,7,8], traveller’s diarrhoea [9] and antibiotic-associated diarrhoea in both children [10,11] and adults [12,13,14]. Prebiotics, usually oligosaccharides or dietary carbohydrates, can act as nutrients for probiotics and may enhance their useful effects when administered together. When incorporated in a single preparation they are known as synbiotics [15].

Recent attention has focused on the capacity of probiotics to modify human immune cell responses and reduce the severity of gastrointestinal infections by bacterial pathogens [16,17,18]. Modulation of gut immunity in this way can alter inflammatory or suppressive immunity and is a key factor for the tolerance or intolerance to antigenic molecules encountered by the mucosa. Fundamental to this process are complex interactions between the mucosal immune apparatus including dendritic cells (DCs), regulatory T (Treg) cells and intestinal bacteria [17]. A proposed mechanism of host immune cell modulation by probiotic bacteria involves a shift in the balance of mucosal responses towards anti-inflammatory cytokine production [19,20]. This effect has been demonstrated for several strains of *Bifidobacterium* which promote the production of the anti-inflammatory cytokine, interleukin (IL) -10 by CD4+ T cells isolated from mouse colon epithelium following oral administration of the probiotic [21]. Further evidence to support the anti-inflammatory properties of probiotics in the gut is provided by in vitro experiments showing an attenuation of lipopolysaccharide (LPS) stimulated inflammatory cytokine production by epithelial cells following pre-incubation with *Lactobacillus reuteri* [22].

DCs are considered the most specialised antigen presenting cells (APCs) within the gut and there is evidence that immunological functions of DCs can be modulated following exposure to probiotics [23]. Co-culture of *Lactobacilli gasseri, johnsonii* and *reuteri* with human myeloid dendritic cells (mDCs) resulted in secretion of IL-12 and IL-18 with skewing of T lymphocyte polarisation towards a T helper (Th) 1 class response [24]. Other studies have demonstrated an induction of the regulatory protein, forkhead box P3 (FoxP3+) T cells and the production of anti-inflammatory cytokines by DCs in response to exposure to probiotics [25]. FoxP3+ T cells are key immune suppressive cells inhibiting the activity of cell mediated immunity. They minimise the function of both CD8+ T cells (generation of antigen-specific cytotoxic T lymphocytes) and CD4+ T cells (production of Th1/Th2 cytokines). They suppress the initiation of T cell interaction with antigens (bacterial/viral) and help to restrict the prominence of tissue immune responses, thereby preventing uncontrolled damage and restoration of immune homeostasis. By inhibiting IL-12 and interferon-gamma (IFN-γ) production they prevent the activation of the innate immune natural killer (NK) cells. Thus, potentially they could be both detrimental and beneficial to the host’s immune defences depending on the pathophysiological state of the gut micro-environment of the host. mDcs in this study were activated and mature, thus increasing their ability to present antigens (bacterial, viral) to CD8+ and CD4+ T cells and enhancing their effector functions. In addition, exposure to probiotic bacteria has been shown to affect DC co-stimulatory molecule expression [26].

Prebiotics are oligosaccharide molecules known to enhance the effects of probiotics. In the rat model, co-administration of a standardized extract of cultured *Lentinula edodes* mycelia (AHCC^®^) with the probiotic *Bifidobacterium longum* BB536 was shown to reduce enzymatic markers of colitis and it is proposed that AHCC^®^ may act synergistically to promote the immunomodulatory properties of Bifidus BB536 although these effects have yet to be studied in human subjects. In the rat model, co-administration of AHCC^®^ with the probiotic *Bifidobacterium longum* (Bifidus BB536) was shown to reduce enzymatic markers of colitis and it is proposed that AHCC^®^ may act synergistically to promote the immunomodulatory properties of Bifidus BB536 although these effects have yet to be studied in human subjects.

Oral immunomodulation is a means for modifying the systemic immune system through an effect on the gut. It is based on a process in which the gut immune system reacts to orally administered antigens to produce proteins that generate systemic adaptive responses [27]. Responses to nutrients, pathogens, commensals, or orally administered antigens to which the gut mucosa is exposed, are dependent on the interaction with mucosal immune cells and lead to subsequent systemic responses [28]. For this reason, the study of blood immune parameters following oral probiotic administration can be reflective of changes that originate at the gut mucosal level. 

This study tested the hypothesis that the oral administration of probiotics and/or prebiotics in healthy volunteers alters blood immune cell phenotypes and aimed to study the effects of orally administered probiotics (Bifidus BB536 (BB536) which contains *Bifidobacterium longum* (Morinaga Milk Industry Co. Ltd., Tokyo, Japan) and/or a standardized extract of cultured *Lentinula edodes* mycelia (AHCC^®^, Amino Up Co. Ltd., Sapporo, Japan), on T cell and DC phenotype and function. An antibiotic treatment phase was included to determine if the immunomodulatory effects of probiotic and prebiotic administration persist following oral antibiotic ingestion.

## 2. Methods

### 2.1. Study Design

In this double-blind, randomised study, four different treatments were administered for a total of 12 days each. After 7 days of baseline treatment, all groups were prescribed a broad-spectrum antibiotic (azithromycin 250 mg once daily) for a total of 5 days (Table 1). 

### 2.2. Inclusion and Exclusion Criteria

Volunteers were eligible to participate if they were male, aged between 18–55 years and able to give informed consent. Volunteers were excluded from the study if they had been diagnosed with diabetes mellitus, any immune disorder or were currently under medical investigation. Exclusions also included steroid medication, smoking/substance abuse, allergy to azithromycin or regular medications, probiotics or nutritional supplements. 

### 2.3. Interventions

Volunteers allocated to the probiotic arm of the study received capsules of BB536, containing 1.25 × 10^10^ CFU/g of the probiotic bacteria, *Bifidobacterium longum.* Those volunteers allocated to the prebiotic arm received capsules containing AHCC^®^. This extract of cultured *Lenitula edodes* mycelia contains polysaccharides, amino acids, and minerals that are utilized by *Bifidobacterium* spp. [29]. Volunteers in the synbiotic arm received both BB536 and AHCC^®^ capsules. Placebos consisted of corn starch within a gelatin capsule at a dose of 200 mg and were identical in appearance to BB536 and AHCC^®^ capsules. Azithromycin (Zithromax, Pfizer Inc., New York, NY, USA), given to all groups after 7 days, consisted of a 250 mg tablet administered in a once daily dose. It has broad spectrum activity against a wide range of bacteria which includes the *Bifidobacterium* spp. [30].

### 2.4. Blood Sampling and Immunological Assays

Healthy volunteers provided a total of 3 blood samples: prior to commencing the study (time point 1), after 7 days of treatment (time point 2) and a final sample at the completion of treatment (day 13, time point 3). Peripheral blood mononuclear cells (PBMCs) were harvested and stored in preparation for immunological assays. A flow diagram summarising the immunological assays is given in Figure 1. A full description of the assay methodology is provided in the Appendix A.

### 2.5. Randomisation, Blinding and Allocation Concealment

Participants were allocated with equal probability to one of the four treatment groups. The randomisation was based on a computer-generated code using random blocks of varying size held on a secure server within the Nottingham Clinical Trials Unit (CTU). Access to the sequence was confined to the CTU data manager. Investigators had access to the treatment allocation for each participant by means of a remote, internet-based randomisation system. The sequence of treatment allocations was concealed until interventions had been assigned and recruitment, data collection, and all other trial-related assessments were complete. Both the subjects and the investigators were blinded to group allocations. The randomisation code was broken only after completion of the statistical analysis. 

### 2.6. Sample Size Calculation

As this was a pilot study, the effect size of the treatments was not known. It was estimated at a sample size of 40 (10 in each) group would help discern whether meaningful differences between the groups could be detected.

### 2.7. Statistical Analysis

Statistical analysis was performed with GraphPad Prism version 8.0.2 for Mac, (GraphPad Software, San Diego, CA, USA, www.graphpad.com) using a repeated measures ANOVA design. Parametric data were analysed with the Student *t* test and non-parametric data with the Wilcoxon test for matched pairs. Differences were considered significant at *p* < 0.05. 

### 2.8. Ethics and Trial Registration

This study was granted approval by the University of Nottingham Medical School Ethics Committee and was registered at http://www.clinicaltrials.gov (NCT01201577).

## 3. Results

### 3.1. Volunteer Baseline Characteristics

There were no significant differences between groups with respect to baseline characteristics of the 40 volunteers studied (Table 2).

### 3.2. Effect of Prebiotic and Probiotic or Antibiotic on Serum Inflammatory Markers

White cell count and C-reactive protein (CRP) concentrations were below 11 × 10^9^/L and 10 mg/L respectively at all time points and there were no statistically significant differences between the four groups.

### 3.3. In Vitro Cytokine Secretion by CD3^+^ T Cells in Response to PDB-Ionomycin

Following in vitro stimulation with phorbol 12,13-dibutyrate (PDB)-ionomycin there was an increase in cytokine secretion from CD3^+^ T cells compared with untreated cells or treatment with dimethyl sulphoxide (DMSO) measured with both cytometric bead array and enzyme-linked immunosorbent sandwich assay. This included IFN-γ (859.5 vs. 70.2 pg/mL, *p* = 0.014) and IL-2 (4418.5 vs. 45.7 pg/mL, *p* = 0.024) (Th1 class) as well as IL-4 (9.96 vs. 2.67 pg/mL, *p* = 0.01), IL-5 (18.78 vs. 8.14 pg/mL, *p* = 0.034), IL-6 (68.88 vs. 13.08 pg/mL, *p* = 0.041) and IL-10 (11.11 vs. 6.18 pg/mL, *p* = 0.029) (Th2 class) This effect was seen for cells in pretreatment and both posttreatment phases and indicates an intact response to stimulation. However, there were no significant differences in the magnitude of cytokine secretion between treatment groups (*p* > 0.05 for all comparisons).

### 3.4. Th1:Th2 Class Cytokine Ratios 

In vitro phorbol 12,13-dibutyrate (PDB)-ionomycin stimulated cytokine release by CD3^+^ T cells categorized according to class demonstrated no shift except when examining the ratio of IFN-γ to IL-4. Following treatment with BB536, there was a significant increase in the ratio of IFN-γ to IL-4 (time point 1 vs. time point 2, *p* = 0.031). However, this ratio returned to baseline levels following the administration of antibiotic (time point 2 vs. time point 3, *p* = 0.047). All other calculated ratios did not demonstrate a shift in Th1 and Th2 class cytokines (*p* > 0.05 for all comparisons).

### 3.5. Foxp3 Expression as a Marker of T Regulatory CELL (Treg) Activity 

Assay of CD4^+^CD25^+^ Foxp3^+^ Tregs by flow cytometry demonstrated a level of <3% of peripheral blood mononuclear cells (PBMCs). Volunteers receiving BB536 demonstrated a significant increase in the level of Foxp3^+^ Tregs (0.45% vs. 1.36%, *p* = 0.046) over the course of treatment. Volunteers receiving AHCC^®^ demonstrated a significant increase in Foxp3^+^ Tregs (0.49% vs. 0.70%, *p* = 0.0059) in the first week of treatment only but the expression of Foxp3^+^ Tregs returned to pretreatment values at the end of the study. There was no difference in Foxp3^+^ Treg expression in volunteers who received either placebo or a combination of BB536 and AHCC^®^ (Figure 2).

### 3.6. Plasmacytoid (pDC) and Myeloid (mDC) Dendritic Cells

Peripheral blood revealed higher cell counts (%) of mDCs compared with pDCs in all treatment groups (Figure 3A i–iv). Between treatment groups, there were no significant differences in the counts of DC subsets except for volunteers receiving the BB536/AHCC^®^ combination. In this group, the proportion of mDCs as a percentage of the total number of DCs was significantly higher at both time points following the commencement of treatment [74.50 (68.25–86.50), *p* = 0.0078 and 80.50 (75.25–84.50), *p* = 0.0059 respectively, Figure 3A iv].

No significant changes in the relative proportions of mDC1 to mDC2 occurred in the treatment groups except for volunteers who received the BB536/AHCC^®^ combination (Figure 3B). In this group, there was a significant increase in the mDC2 subset as a percentage of total mDCs at both post-treatment time points [0.149 (0.140–0.174), *p* = 0.002 and 0.138 (0.122–0.162) *p* = 0.002, Figure 3B iv].

### 3.7. Co-Stimulatory Molecule Expression in DC Subsets

There were no differences between CD40 and CD86 surface expression on pDCs and mDCs between treatment groups (*p* > 0.05 for all comparisons). (Figure 4A).

However, there was a significant increase in both CD40 (time point 1 vs. time point 2: 18.5% vs. 30.5%, *p* = 0.029 and time point 1 vs. time point 3: 18.5% vs. 36.5%, *p* = 0.027, Figure 4A iv) and CD86 (time point 1 vs. time point 2: 24% vs. 38%, *p* = 0.034 and time point 1 vs. time point 3: 24% vs. 40.5%, *p* = 0.029, Figure 4B iv) surface marker expression in mDC2 subset for volunteers receiving the BB536 and AHCC^®^ combination.

## 4. Discussion

In this study of healthy volunteers, we have demonstrated an increase in the percentage of mDCs following the oral administration of a combination of BB536 and AHCC^®^. This is in contrast to the percentage of pDCs. Treatment with this combination also led to a shift towards the mDC2 subset with greater expression of the co-stimulatory molecules CD40 and CD86, suggesting DC activation and maturation, albeit mDC2 is the less common subset.

On the other hand, treatment with either BB536 or AHCC^®^ did not lead to any differences in PDB-ionomycin stimulated cytokine release from CD3^+^ T cells. Foxp3^+^ Tregs, however, were increased with either BB536 or AHCC^®^ but not in combination.

The results of the present study suggest differential expansion and activation of DC subsets in response to the oral administration of a synbiotic preparation. This effect was observed to persist even after the administration of azithromycin 250 mg daily for 5 days. In the gut, both mDCs and pDCs are responsible for presenting microbial and dietary antigens to the adaptive immune system, thereby influencing polarisation of the adaptive response via cytokine and metabolite production [31,32]. The results of the present study indicate that oral ingestion of synbiotic preparations may modulate DC subgroups and promote expansion of the less common mDC2 subset. This in turn would favour a Th2 type of cytokine response and suppression of inflammatory cytokine secretion. This, however, was not borne out by our observations for stimulated T lymphocyte helper responses which remained similar for all treatment groups. 

Previous work has indicated that probiotic bacteria can influence cytokine release from different classes of immune cells. However, much of this work has been conducted in animals or the in vitro setting mainly examining effects of lactic acid-producing bacteria [33,34,35,36,37]. When examining specifically the role of T lymphocytes, most studies have investigated the direct effects of co-culturing probiotic bacteria with PBMCs [38,39,40,41]. A further study examined cytokine production by human T lymphocytes following co-culture with various strains of bacteria including *Bifidobacterium lactis*, *Lactobacillus acidophilus*, *Lactobacillus plantarum* and *Escherichia coli* [39]. They were able to demonstrate increased production of IL-10, IL-17, IFN-γ and IL-13, although there were no significant differences in cytokine production between strains [39]. IL-10 is known to inhibit Th1 proliferation and suppress IFN-γ and IL-2 release. A possible mechanism for probiotic action could involve the induction of IL-10 release from T lymphocytes and this has been shown for colonic lamina propria and Peyer’s patch-derived T lymphocytes by *Bifidobacterium* strains in mice [21,42]. In the present study, concentrations of IL-10 following stimulation of T lymphocytes with PDB-ionomycin were not significantly elevated above those obtained with controls. There were also no significant differences in IL-10 concentrations between groups receiving placebo, BB536, AHCC^®^ or combination. A healthy human volunteer study investigating cytokine release following treatment for 8 weeks of *Bifidobacterium infantis* and stimulation of PBMCs found increased IL-10 production but no difference in the production of IL-2, IL-12p70, tumour necrosis factor-alpha (TNF-α) and IFN-γ [25]. No studies have examined the optimum treatment duration required to observe changes in immune cell cytokine production. It is, therefore, possible that the treatment duration in the present study may have been insufficient to produce overt changes in cytokine release.

In the present study, volunteers receiving BB536 demonstrated a significant increase in the percentage of CD4^+^ lymphocytes expressing Foxp3 following the total treatment period, an effect not observed for volunteers receiving placebo, AHCC^®^ or combination treatment. Probiotic bacteria have been shown in several studies to increase the expression of Foxp3. Cells expressing Foxp3 are known to play an important suppressive role in the immune system by directly modulating the expansion and function of T-cells. Again, these experiments have mainly utilised probiotic strains co-cultured with PBMCs in vitro or have been conducted in murine studies [21,37,39,41]. On the whole, these studies indicate that probiotic bacteria and *Lactobacillus* spp. in particular induce the expression of Foxp3 from PBMCs and, in addition, induce the expression of immunosuppressive cytokines, such as IL-10 [21,39]. A study involving healthy human volunteers treated with *Bifidobacterium infantis* for 8 weeks demonstrated a significant increase in the percentage of CD4^+^ CD25^high^ T cells that also expressed Foxp3 [25]. The investigators subsequently co-cultured *Bifidobacterium infantis* with monocyte-derived DCs which were shown to increase expression of Foxp3 and production of IL-10. They were also able to demonstrate differential mechanisms for Foxp3 induction by DCs, with an mDC pathway requiring TLR-2, DC-SIGN and retinoic acid and a pDC pathway requiring indolamine 2,3-dioxygenase (IDO) [25]. In the present study, an increase in CD4^+^ lymphocytes expressing Foxp3 following treatment with BB536 was not observed with combination treatment of BB536 and AHCC^®^. Oligosaccharides such as AHCC^®^ are known to promote proliferation of *Bifidobacterium* although it remains unclear how long this positive interaction lasts or to what degree intestinal colonisation proceeds as a result. One could speculate that the effects on Tregs following combination treatment are population dependent and an increase in BB536 intestinal population beyond a specific threshold could result in attenuation of this effect. It is also possible that by favouring the proliferation of BB536, an adhesive or colonising phenotype might be prevented when a preferred dietary substrate is present. This may account for the lack effect seen when administering the two components together.

The polarisation of host immune responses and the effects on DCs, T helper and regulatory cells are likely to be central to the observed effects of probiotic administration when given for therapy in gastrointestinal infections. Mechanisms for this seem centred on the induction of Tregs, modulation of cytokine expression or differential activation of DC subsets. This has been specifically demonstrated for *Lactobacillus* spp. with a differential induction of IL-12 and TNFα [43]. Others have shown augmented expression of the DC maturation markers, MHC II and CD86 [44]. There is evidence for polarised cytokine DC subset activation in response to helminth exposure with a predominance for Th2 cytokine expression for activated cells [45]. Similarly, Polysaccharide A (PSA), the archetypical immunomodulatory molecule of the gut commensal *Bacteroides fragilis*, induces Tregs to secrete the anti-inflammatory cytokine IL-10 and *Bifidobacterium bifidum* has been shown to be an effective inducer of Tregs in response to dietary antigens [46,47]. Results from recent human studies indicate similar anti-inflammatory mechanisms in regard to cytokine expression and effector cell phenotypes. Analysis of LPS stimulated PBMCs obtained from human volunteers following ingestion of *Bifidobacterium animalis* subsp. *lactis* BB-12 demonstrated lower TNF-α secretion and reduced expression of TLR-2 on CD14^+^HLA-DR^+^ cells [48]. A randomised human study of patients with metabolic syndrome demonstrated a reduction in plasma concentrations of both IL-6 and TNF-α following 45 days of oral treatment with fermented milk containing *Bifidobacterium lactis* [49]. In this way probiotic bacteria can enhance the mucosal immune system to control responses to luminal antigens and limit mucosal inflammation which is a central pathogenic feature of gastrointestinal infections.

There are a number of limitations of this study which should be considered. Firstly, this study was conducted with healthy volunteers, in whom responses both immunological and physiological are likely to be different from those with illness. It could be argued that the observed changes in immune cell phenotype and function may not be directly applicable to clinical settings of physiological stress, such as surgical trauma or sepsis. However, the intention of this study was to observe the effects of probiotic and prebiotic ingestion in health to determine immunological effects in the absence of other confounding factors such as sepsis, trauma or systemic inflammation. Differential expansion of DC subsets persisted even after antibiotic treatment. There is some evidence in the mouse model that pasteurized probiotic bacteria can still affect metabolic processes and it may have been of value in the present study to include a heat-inactivated control to determine if the immunomodulatory properties are seen with inactivated bacteria [50]. It could also be argued that the study did not incorporate an objective biological measure of treatment compliance. Collectively, volunteers tolerated the treatments well and there were no significant adverse effects associated with any of the treatment groups. In addition, at each sampling time point volunteers were asked to bring the medications prescribed and remaining capsules were counted. Nevertheless, the risk of non-compliance was significant and remains a possible limitation of the study. In an effort to limit the effects of cyclical variation of endogenous steroid hormones on cellular immunity we recruited only male subjects to participate in this study. It is acknowledged that this may limit the wider extrapolation of these results. Finally, whilst this study has demonstrated immunomodulation following synbiotic ingestion, the pathways responsible for immunomodulation are clearly complex and warrant further investigation to elucidate precise mechanisms. 

To summarise, the results obtained in this study demonstrate that in healthy individuals, oral supplementation with BB536 and AHCC^®^ or combinations had varying effects on T cells and dendritic cell populations and the expression of surface markers. The administration of either BB536 or AHCC^®^ was associated with increased expression of Foxp3 by T cells. The synbiotic combination of BB536 and AHCC^®^ was associated with a preferential increase in mDCs, and in particular the mDC2 subset. In addition, combination treatment was associated with increased expression of activation markers CD40 and CD86. These immunomodulatory effects warrant further investigation to uncover the mechanisms by which probiotics/prebiotics may exert any beneficial effects.

## Figures and Tables

**Figure 1 nutrients-11-02470-f001:**
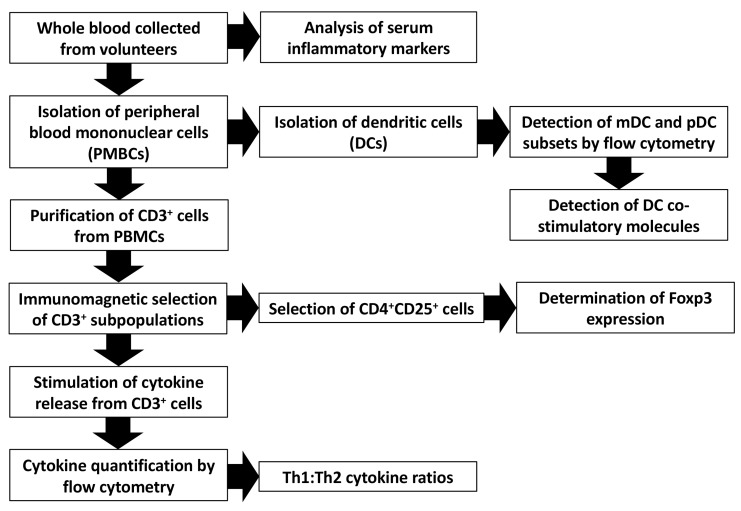
Outline of methods used to isolate immune cells and cytokines. Foxp3: forkhead box P3; mDC: myeloid dendritic cells; pDC: plasmacytoid dendritic cells; Th: T helper.

**Figure 2 nutrients-11-02470-f002:**
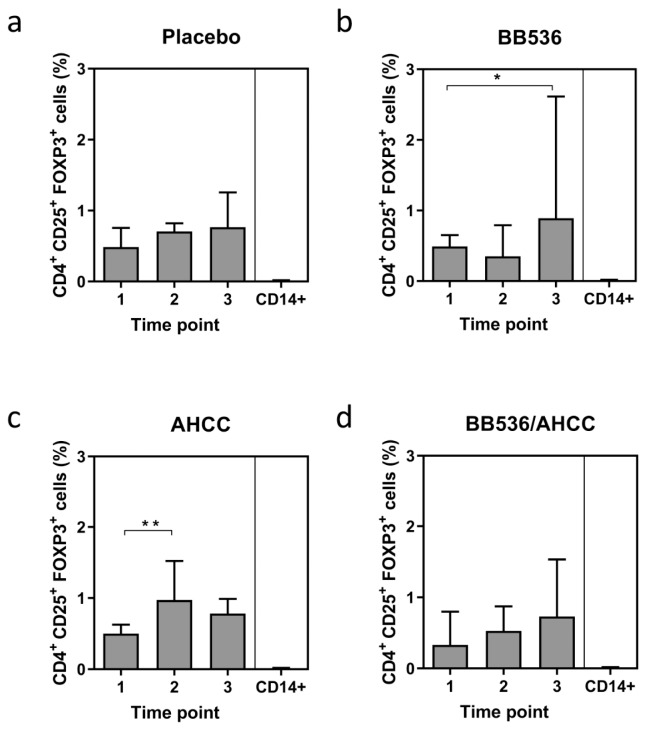
CD4^+^ CD25^+^ Foxp3^+^ cells (Tregs) as determined by cytometric bead array. X-axis represents different sampling time points during the study. Values represent mean (SEM). Significance calculated using the Student *t* test (paired). BB536: *Bifidobacterium longum*; AHCC^®^: a standardized extract of cultured *Lentinula edodes* mycelia; *: *p* < 0.05; **: *p* < 0.01. Time points: 1—pretreatment, 2—after 7 days of placebo (**a**)/BB536 (**b**)/AHCC^®^ (**c**)/both (**d**), 3—after an additional 5 days of placebo (**a**)/BB536 (**b**)/AHCC^®^ (**c**)/both (**d**) + azithromycin (day 13).

**Figure 3 nutrients-11-02470-f003:**
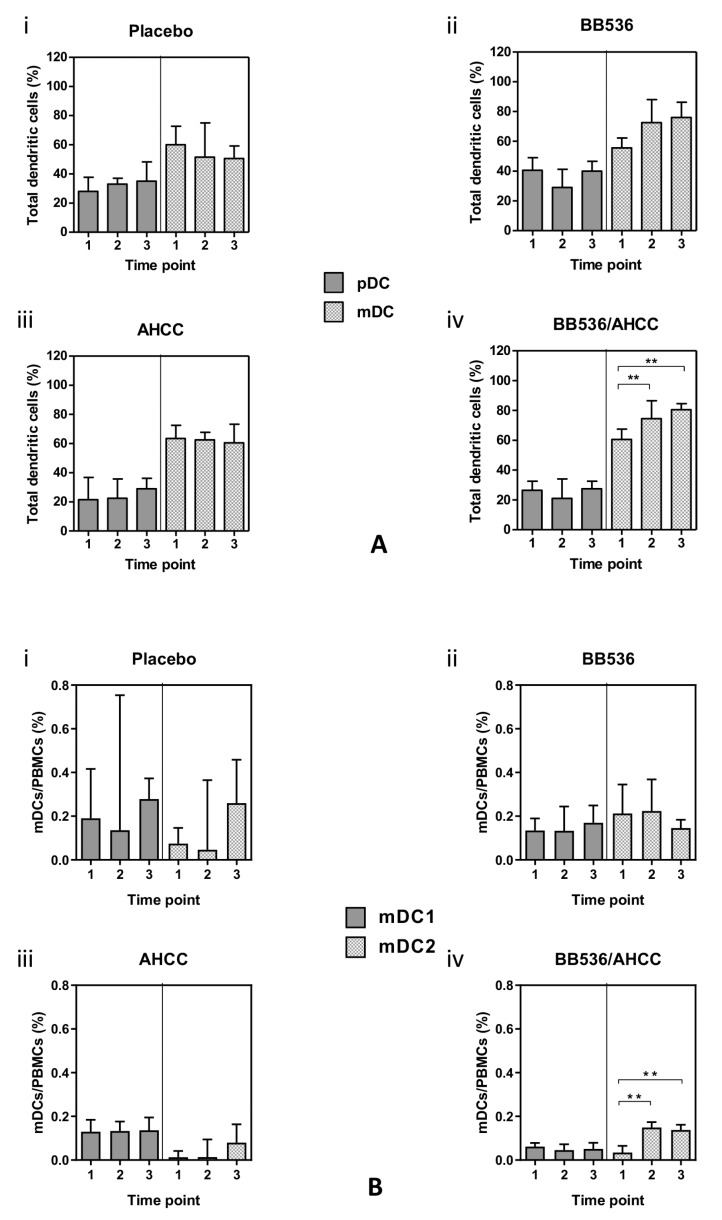
(**A**) Characterisation of DC subsets based on expression of CD11c as a percentage of total mDCs. (**B**) Characterisation of myeloid dendritic cell subsets as a percentage of total PMBCs. Values represent mean (SEM). Significance calculated using the Student *t* test (paired). BB536: *Bifidobacterium longum*; AHCC^®^: a standardized extract of cultured *Lentinula edodes* mycelia; **: *p* < 0.01. Time points: 1—pretreatment, 2—after 7 days of placebo (i)/BB536 (ii)/AHCC^®^ (iii)/both (iv), 3—after an additional 5 days of placebo (i)/BB536 (ii)/AHCC^®^ (iii)/both (iv) + azithromycin (day 13).

**Figure 4 nutrients-11-02470-f004:**
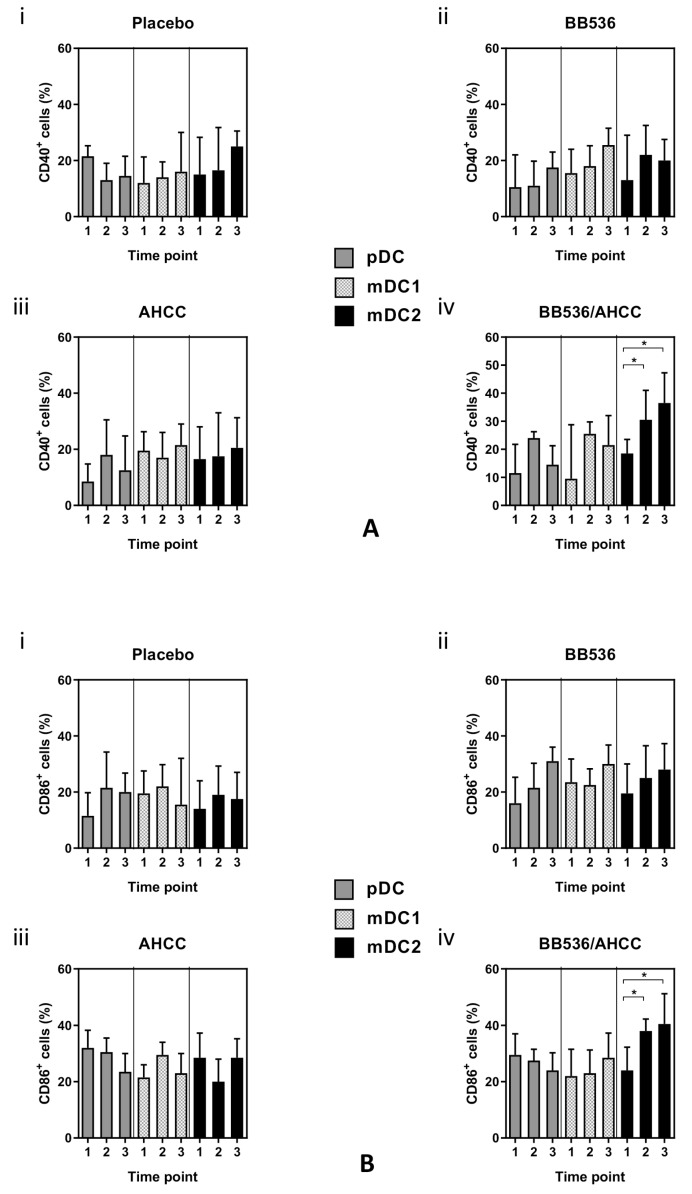
(**A**) Cell surface co-stimulatory marker CD40 expression on DC subsets as percentage of total DCs. (**B**) Cell surface co-stimulatory marker CD86 expression on DC subsets as percentage of total DCs. Values represent mean (SEM). Significance calculated using the Student *t* test (paired). BB536: *Bifidobacterium longum*; AHCC^®^: a standardized extract of cultured *Lentinula edodes* mycelia; *: *p* < 0.05. Time points: 1—pretreatment, 2—after 7 days of placebo (i)/BB536 (ii)/AHCC^®^ (iii)/both (iv), 3—after an additional 5 days of placebo (i)/BB536 (ii)/AHCC^®^ (iii)/both (iv) + azithromycin (day 13).

**Table 1 nutrients-11-02470-t001:** Summary of healthy volunteer randomisation and treatment regimens.

Group	Treatment Length	Intervention
1 (*n* = 10) Placebo arm	7 days	Placebo prebiotic and placebo probiotic
	5 days	Azithromycin and placebo prebiotic and placebo probiotic
2 (*n* = 10) Probiotic arm	7 days	Placebo prebiotic and BB536
	5 days	Azithromycin, placebo prebiotic and BB536
3 (*n* = 10) Prebiotic arm	7 days	AHCC^®^ and placebo probiotic
	5 days	Azithromycin, AHCC^®^ and placebo probiotic
4 (*n* = 10) Synbiotic arm	7 days	AHCC^®^ and BB536
	5 days	Azithromycin, AHCC^®^ and BB536

AHCC^®^, a standardized extract of cultured *Lenitula edodes* mycelia (Amino Up Co. Ltd., Sapporo, Japan) 300 mg in a gelatin capsule. *Bifidobacterium longum* (BB536, Morinaga Milk Industry Co. Ltd., Tokyo, Japan) 1.25 × 10^10^ CFU/g in a gelatin capsule. Placebos: Corn starch 200 mg in a gelatin capsule identical to AHCC^®^ and BB536 capsules. Antibiotic: Azithromycin dehydrate (Zithromax, Pfizer Inc., New York, NY, USA) 250 mg tablet given once daily.

**Table 2 nutrients-11-02470-t002:** Baseline characteristics of volunteers in the 4 treatment groups. Values presented are mean (SEM).

	Normal Range	Placebo	BB536	AHCC^®^	BB536/AHCC^®^
Age (yr)		20.3 (0.2)	21.1 (0.2)	20.9 (0.3)	22.5 (0.3)
Height (m)		1.82 (0.02)	1.77 (0.02)	1.78 (0.02)	1.80 (0.02)
Weight (kg)		63.1 (0.3)	65.4 (0.2)	77.2 (0.3)	68.9 (0.2)
Haemoglobin (g/dL)	130–180 g/L	14.1 (0.2)	15.0 (0.2)	13.9 (0.2)	14.8 (0.2)
White cell count (×10^9^/L)	4.0–11.0 × 10^9^/L	6.9 (0.7)	8.7 (0.5)	6.8 (0.7)	8.1 (0.5)
Platelet count (×10^9^/L)	140–400 × 10^9^/L	302 (8)	323 (7)	298 (7)	335 (8)
Serum bilirubin (μmol/L)	<21 μmol/L	10 (2)	8 (1)	14 (2)	6 (1)
Serum alanine amino transferase (IU/L)	2–53 iu/L	25 (4)	27 (3)	30 (4)	19 (3)
Serum alkaline phosphatase (IU/L)	30–130 iu/L	58 (5)	77 (6)	63 (5)	71 (5)
Serum albumin (g/L)	35–50 g/L	38 (0.4)	39 (0.4)	36 (0.4)	41 (0.4)
Serum sodium (mmol/L)	133–146mmol/L	134 (0.5)	134 (0.4)	135 (0.4)	133 (0.5)
Serum potassium (mmol/L)	3.3–5.3mmol/L	4.8 (0.2)	4.2 (0.2)	4.2 (0.2)	4.5 (0.2)
Blood urea (mmol/L)	2.5–7.8 mmol/L	5.2 (0.2)	4.5(0.2)	3.6 (0.2)	4.3 (0.2)
Serum creatinine (mmol/L)	60–120μmol/L	98 (3)	77 (3)	64 (2)	74 (3)
Serum C-reactive protein (mg/L)	0–10 mg/L	8.1 (0.5)	8 (0.4)	7 (0.4)	5 (0.4)

There were no statistically significant differences in baseline parameters between the groups (*p* > 0.05).

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
