# Peer review of "Modulation of T Regulatory and Dendritic Cell Phenotypes Following Ingestion of Bifidobacterium longum, AHCC® and Azithromycin in Healthy Individuals"

_nutrients, 2019, doi:10.3390/nu11102470_

Round 1
Reviewer 1 Report
This is an interesting article and has potential to contribute an update of knowledge in the field. The content of this article is fully consistent with its title. Presented a problem developed on the basis of current knowledge. Using appropriate analytical methods. The scope of the analysis and interpretation of test results is correct. Results and conclusions represent a contribution to the development of this field of knowledge. The subject of this article is important. Used the proper scope of the literature.
The idea of this manuscript is good and well written but there are some negative points such as:
The methodology does not provide a detailed description of the research methods. Which reagents were used, in what quantities, the conditions of analysis, the apparatus used. If this is described in other articles, please refer to this research.
Too short, general description of the results.
In the description under the charts there is no information as to which statistical test was used. In the methodology, in the specification of tests, the 0.05 confidence interval was given, and 0.01 in the case of extrinsic dashes.
Poor quality of charts 3-4 and especially 4.
Reviewer 2 Report
Brief Summary:
The current study investigates the use of probiotics, prebiotics, or a combination in healthy human volunteers to assess their immunomodulatory impact on systemic immune cells (e.g. T-cells, PBMCs, and DCs). Following 7 days of treatment, antibiotics are then supplemented to the treatments. The paper demonstrates supplementation with a probiotic or synbiotic may increase the expression of immune cell markers or mDCs.
Broad Comments:
A weakness of the paper lies in the introduction, as it does not adequately address the importance of why they are including a period of antibiotic use. Is post-antibiotic treatment immunomodulation a good thing?
The novelty of your manuscript lies in the fact that it is a human study, and it will determine the impact of prophylactic / concomitant probiotic/prebiotic treatment on the adverse effects of antibiotics, yet this isn’t adequately emphasized.
Without a better defined objective of the study, I find the administration of antibiotics to 40 healthy individuals irresponsible, as it is found that many bacterial species will not re-colonise the host, and this also contributes to overuse of antibiotics.
Also missing is the importance of anticipated / observed results. Why are changes in activation markers / Foxp3 / mDCs interesting / important? As such, the introduction/discussion is lacking.
A heat-inactivated bacterial control would have helped, as administration of probiotics in tandem with antibiotics (which were mentioned in the paper to be active against bifidobacteria) appears to be counterintuitive. There is a recent study of Akkermansia supplementation which found that pasteurized bacteria displayed a significant effect. Thus, it is not clear if dead bacteria would have the same impact as live bacteria, and may explain why the immunomodulatory effect persists even following antibiotic treatment.
Lack of any mechanistic study in the paper limits its impact and interpretation.
Additionally, the fact that the subjects were only male will limit the extrapolation of the results.
A strength of the paper lies in acknowledging its limitations. These include low subject numbers, short-duration of treatment, and possible treatment non-compliance. A further strength is the fact that it was double-blind.
Specific Comments:
21: ..which contains Bifidobacterium longum, has been [delete the ‘and’]
22: add ‘s’ to effect: “…shown to have enhanced probiotic effects..”
30: ‘Cytokine profiles from purified CD3+ cells’ should be changed to ‘CD3+ T cells’
34-35: The sentence is confusing. There was a change in Foxp3 and IFN-g/IL-4 ratios in those receiving BB536, yet there were no significant differences between groups. Please clarify.
36: Are the percentages reversed? i.e. increase in myeloid versus plasmacytoid DC counts (61% vs. 80%) at post-treatment timepoints. Please clarify.
37: Still having trouble grasping the data. mDC2 phenotypes were more prevalent following combination treatment (0.05% vs. 0.16%).. is this baseline versus day 7/12? Please clarify.
44-47: While the paragraph introduces potential downsides to antibiotic use, it should do a better job to link to the following paragraph. I find that each paragraph in the introduction begins and finishes fairly abruptly, disrupting the flow / readability for the reader. An example linking statement: To reduce or eliminate these adverse effects, probiotic treatment has been previously investigated… and then you can continue to describe what probiotics are and how they are helpful.
63: please elaborate / give examples of anti-inflammatory cytokine production.
64: Please indicate the model in which these results were demonstrated. Too vague.
66-68: Probiotic pre-incubation with epithelial cells? Co-administration? Etc, as exposure to E. coli in the presence of L. reuteri could just imply that L. reuteri may inhibit the interaction of E. coli with the epithelial cells, rather than L. reuteri modulating the immune response via the epithelial cells.
68: repetition of the word ‘probiotic’
71: ‘Lactobacilli’ is too vague, was this a general effect of all lactobacilli?
73: ..forkhead box P3 (Foxp3)-positive T cells? (or are they Foxp3 T cells?)
73: What is the importance of Foxp3, is it associated with an anti-inflammatory cytokine secretion?
74: I am not sure ‘immunosuppressive’ is appropriate here.
81: Insert comma: ‘..exposed, are..’
85: Tense: .. alters, rather than altered.
86-92: The introduction did not introduce the two main treatment components of the study. Why were these particular components used? Should outline any previous in vitro results, as this will help to shape the present hypothesis.
91-92: Very vague, what is the goal? To determine the persistence of any changes? The effects of probiotic/prebiotic supplementation on intestinal microbiota recovery?, etc..
105: Please comment on the inclusion criteria: Women were not included, yet men could cover a 37 year age range? Was BMI not taken into account? This could have a large impact on the eventual changes in immune response / systemic inflammation.
114: Repetition: ‘This extract extract…’
118: Would the same dose of antibiotic have the same impact on all study participants?
122-123: Was sampling done at the same time each day? Would this impact any immune cells / activity found?
140-142: Thank you for including this explanation.
146: The ‘t’ in Student’s t-test should be italicized and it is ‘Student’s’, as I believe this is someone’s last name.
153: I find it hard to believe that subjects were randomly assigned to groups, with one group (BB536) having a mean weight of 85.4kg while the others are 63, 77, and 68.9, with such small SEM. Were there really no statistically significant differences between baseline parameters between groups? I’d think this difference in weight alone would potentially impact probiotic / antibiotic activity.
158: Please define ‘normal limits’
162-164: Does this include all groups? Please clarify.
175: What is the increase?
191-194: This section could use a bit of clarification.
196: increase in percentage of mDCs.. in relation to what?
200: Should this not be emphasized? The treatment has managed to enrich a less common subset of mDCs, is this not of interest?
202: Please be consistent throughout the manuscript, this is one of the rare instances where it is explicitly written as CD3+ T cells.
209-211: Are the problems following antibiotic administration related to inflammation? Or is it the re-colonisation by pathogens/microbes which drives inflammation? Would limiting the immune response during pathogen re-colonisation be a good thing?
214-234: This section is well written. You present a potential mechanism, followed by examples for the rationale and how your study may be different (in vitro / mice versus human studies), and finally, your results, how they fit with what has been seen before, and a potential explanation. Excellent.
228: Reverse word order: A healthy human volunteer.
235-237: Those receiving BB536 saw an increase, but not those receiving placebo, what about the other 2 groups?
238: The importance of Foxp3 hasn’t been clearly introduced (if it has, my apologies, I missed it). You could include a modified version of “Plays an essential role in maintaining homeostasis of the immune system by allowing the acquisition of full suppressive function and stability of the Treg lineage, and by directly modulating the expansion and function of conventional T-cells.” which I’ve taken directly from the UniProt website.
Again, I think you need to emphasize why you’d like to see immunosuppression during antibiotic use, or to say that you’d like to see if the resulting effects would persist even following the removal of the bacteria / microbiome through antibiotic use.
239: I would suggest using ‘Together / Collectively’ as opposed to ‘On the whole’, but not overly important as something to change.
251-253: Please clarify the speculation. Do suggest that surpassing a threshold bacterial load would attenuate the positive effects and trigger an inflammatory response?
My opinion is that likely the BB536 is digesting the prebiotic, thus removing its availability from other strains in the gut. At the same time, bifidobacteria activity can differ based on metabolism, and will not display an adhesive/colonising phenotype when a preferred dietary substrate is present (e.g. lactose). This may account for the lack effect seen when administering the two components together?
255: It’s not necessarily about the host immune response, but more of the probiotics competing for space against the pathogens, whether by occupying a niche or direct competition with pathogens by the release of bacteriocins, etc.. I think you’re speculating a bit too much, as the examples given after this statement don’t seem to provide any direct evidence to support it.
269: What type of cells? T cells?
275:Good that you’ve addressed this aspect.
286: If the treatment was well-tolerated, there wouldn’t be a significant risk of non-compliance, other than human forgetfulness.
289: The summary is a bit overly vague, as “..may have effects on innate and adaptive immune…”. If you’ve demonstrated an effect, say so. Prebiotic/Probiotic treatment had varying effects on the populations of cells or expression of surface markers.. and then you can proceed to reiterate them if you’d like.
Supplemental Data.
54-62: Way too much detail. ‘Assay reagents were prepared according to manufacturer’s instructions.’
73: Not necessary to indicate what the tube was labelled.
86: Were there any group effects seen from the samples stored for 18 months? Can these cells be stored this long? Please comment in this section to clarify.
120: 100 ul were distributed per well in duplicate (2 plates? 2 wells each sample?). This isn’t very clear / relevant.
127: anti-human.
Reviewer 3 Report
The authors studied the effects of orally administered Bifidus BB536 which contains Bifidobacterium longum and/or active hexose correlated compound on T cell and DC phenotype and function, and also determined the effect of oral administration of azithromycin on these changes. They reported some interesting findings that oral intake of AHCC and BB536 may modulate T regulatory and DC phenotypes to favour anti-inflammatory responses following antibiotic usage.
The title is so general and broad, and cannot reflect what you did in this study. A more specific title covering the major objectives of this study should be used.
The description in Results were so brief and simple. You should definitely add more details to it.
The data should be described more statistically in Results. Line 153: If there were no significant differences, you should add P>0.05. Line 161-162: Was the increase significant? If yes, you should add P<0.05. You should correct this in Results.
Round 2
Reviewer 3 Report
Some of the comments have not been addressed. The authors should write a response to the reviewer's previous comments:
The authors studied the effects of orally administered Bifidus BB536 which contains Bifidobacterium longum and/or active hexose correlated compound on T cell and DC phenotype and function, and also determined the effect of oral administration of azithromycin on these changes. They reported some interesting findings that oral intake of AHCC and BB536 may modulate T regulatory and DC phenotypes to favour anti-inflammatory responses following antibiotic usage.
The title is so general and broad, and cannot reflect what you did in this study. A more specific title covering the major objectives of this study should be used.
The description in Results were so brief and simple. You should definitely add more details to it.
The data should be described more statistically in Results. Line 153: If there were no significant differences, you should add P>0.05. Line 161-162: Was the increase significant? If yes, you should add P<0.05. You should correct this in Results.
